# Electrophysiological Impact of SARS-CoV-2 Envelope Protein in U251 Human Glioblastoma Cells: Possible Implications in Gliomagenesis?

**DOI:** 10.3390/ijms25126669

**Published:** 2024-06-18

**Authors:** Lorenzo Monarca, Francesco Ragonese, Andrea Biagini, Paola Sabbatini, Matteo Pacini, Alessandro Zucchi, Roberta Spaccapelo, Paola Ferrari, Andrea Nicolini, Bernard Fioretti

**Affiliations:** 1Department of Chemistry, Biology and Biotechnologies, University of Perugia, 06123 Perugia, Italy; lorenzo.monarca@unipg.it (L.M.); francesco.ragonese@unipg.it (F.R.); andrea.biagini@dottorandi.unipg.it (A.B.); paola.sabbatini@unipg.it (P.S.); 2Department of Medicine and Surgery, Perugia Medical School, University of Perugia, 06132 Perugia, Italy; roberta.spaccapelo@unipg.it; 3Urology Unit, Department of Translational Research and New Technologies in Medicine and Surgery, University of Pisa, 56126 Pisa, Italy; matteopacini93@gmail.com (M.P.); alessandro.zucchi@unipi.it (A.Z.); 4Interuniversity Consortium for Biotechnology (C.I.B.), 34148 Trieste, Italy; 5Department of Oncology, Transplantations and New Technologies in Medicine, University of Pisa, 56126 Pisa, Italy; p.ferrari@ao-pisa.toscana.it

**Keywords:** SARS-CoV-2, E protein, glioblastoma, mitochondria depolarization, cell proliferation

## Abstract

SARS-CoV-2 is the causative agent of the COVID-19 pandemic, the acute respiratory disease which, so far, has led to over 7 million deaths. There are several symptoms associated with SARS-CoV-2 infections which include neurological and psychiatric disorders, at least in the case of pre-Omicron variants. SARS-CoV-2 infection can also promote the onset of glioblastoma in patients without prior malignancies. In this study, we focused on the Envelope protein codified by the virus genome, which acts as viroporin and that is reported to be central for virus propagation. In particular, we characterized the electrophysiological profile of E-protein transfected U251 and HEK293 cells through the patch-clamp technique and FURA-2 measurements. Specifically, we observed an increase in the voltage-dependent (Kv) and calcium-dependent (KCa) potassium currents in HEK293 and U251 cell lines, respectively. Interestingly, in both cellular models, we observed a depolarization of the mitochondrial membrane potential in accordance with an alteration of U251 cell growth. We, therefore, investigated the transcriptional effect of E protein on the signaling pathways and found several gene alterations associated with apoptosis, cytokines and WNT pathways. The electrophysiological and transcriptional changes observed after E protein expression could explain the impact of SARS-CoV-2 infection on gliomagenesis.

## 1. Introduction

SARS-CoV-2 is the causative agent of the COVID-19 pandemic, the acute respiratory disease that has been spread all over the world since 2019, with over 7 million of deaths reported by the World Health Organization [1]. Several approaches, including multiple therapeutic methods, have been used in order to prevent virus infection, such as vaccines and small molecules, as well as to reduce the severity and the mortality associated with COVID-19 manifestation [2]. Most of these treatments target the viral spike S-protein, which is responsible for binding to Angiotensin Converting Enzyme 2 (ACE2) and for the following fusion and entrance of the virus in the cell. Currently, several mutations of the SARS-CoV-2 virus have been reported (D614G, the Alpha variant, the Beta variant, the Gamma variant, the Delta variant, and the Omicron variant, which diversified into numerous subvariants), with different responses to vaccines and antibodies [3]. The symptoms associated with SARS-CoV-2 infections are varied and include not only fever, cough, and pulmonary diseases, but also gastrointestinal ailments and, at least in the case of pre-Omicron variants, neurological diseases, such as muscle soreness, headaches, dizziness, altered smell and taste as well as confusion and even psychiatric disorders [4]. Neurological symptoms are also a characteristic of the long COVID, a multisystemic syndrome that persists beyond the acute phase of the infection [5]. A clinical retrospective study reported that one-third of COVID-19 survivors presents neurological or psychiatric symptoms six months after the infection [6].

A statistical analysis performed in a large cohort of European people affected by SARS-CoV-2 infection demonstrated that genetically predicted COVID-19 hospitalization can be significantly considered a promoting factor for the onset of glioblastoma in patients without prior malignancies [7]. SARS-CoV-2 replication was also observed in some glioblastoma cell lines, where it can also represent a reservoir for the virus in the body [8]. Moreover, it has been recently reported that SARS-CoV-2 causes an excessive inflammatory response that leads to increased permeability of the blood–brain barrier, endothelial dysfunctions, and it can significatively impact glioblastoma progression [9]. SARS-CoV-2 infection can also activate the transcription of the Ephrin (Eph) receptors that, in the brain cells, are responsible for COVID-19-related neurological disorders and for the progression of neurodegenerative diseases like Alzheimer’s disease and cancers such as glioblastoma [10].

In 2020, Chu et al. suggested a pathogenetic mechanism for COVID-19 symptomatology analyzing cell susceptibility, species tropism, replication kinetics, and virus-induced cell damage in nine human cell lines, demonstrating that SARS-CoV-2 is able to replicate also in U251 glioblastoma cells, even if only in a modest amount [11]. This replication can be causative of the neurological manifestations that are present in COVID-19 patients as confusion, anosmia and ageusia. It was also demonstrated that SARS-CoV-2 can infect other neuroblastoma and glioblastoma cell lines (SH-SY5Y, SK-N-BE, U87 and U373) showing a trophism for neuronal cells, even if no replication or cytopathic effect was observed [12]. As a support of these findings, the presence of crucial SARS-CoV-2 entry factors, in particular of neuropilin-1, was found in surgically derived glioblastoma tissues and cell lines [13], and in vitro studies showed a kind of trophism of the virus for a variety of neuronal and glial cell types [14].

Among the four structural proteins that are codified in SARS-CoV-2 genome, we focused our attention on envelope (E) protein, a 75 amino acid-long protein that is well conserved in *Coronavirus* family. There is a high homology in the sequence alignment of SARS and SARS-CoV-2, which allowed us to infer a similarity also in the functions and the behavior of the two proteins [15]. E protein consists of a hydrophobic transmembrane domain (TMD) and two amphiphilic α-helices H2 and H3, connected by flexible linkers. When immersed in a lipid bilayer, E protein forms a homopentameric ion channel structure and its transmembrane domain forms a five-helix bundle surrounding a narrow pore [16]. It is known that the deletion of the E gene from viral genomes results in an attenuate virion [17] since it plays pivotal roles in virus assembly, budding, release, inflammasome activation and virus propagation. Double immunofluorescence analysis showed that E protein is mostly localized in the endoplasmic reticulum and in the Golgi apparatus of the host cell [18]. There are several issues regarding the membrane incorporation of the E protein and the plasma membrane current associated with its expression in different kinds of model cells, and the activity of E protein as an ion channel is still not fully understood [19]. While some authors showed a cationic channel activity for E protein [20], recently, it was hypothesized that the recorded currents are connected to the induction of endogenous pannexin channels [21].

The previously reported evidence of the impact of SARS-CoV-2 replication on neuronal and glial cell lines and our interest in the field of glioblastoma treatment [22,23,24,25] prompted us to study the electrophysiological and pathophysiological profile of immortalized glioblastoma cells (U251) expressing SARS-CoV-2 virus E protein. U251 cells were selected as the cell line on which to perform the studies because they represent a robust and established model of human glioblastoma [26]. Moreover, literature data suggest that SARS-CoV-2 is able to replicate in this cell line [8]. From an electrophysiological viewpoint, U251 is also well characterized with the expression of intermediate conductance (IKCa, Iuphar name KCa3.1) and big conductance (BKCa, Iuphar name KCa1.1) calcium-activated and inward rectifier (KIR, Iuphar name Kir4.1) potassium currents [27,28,29] and cationic currents [24].

In this study, we characterized the electrophysiological, proliferative and metabolic effects induced by E protein expression in the U251 glioblastoma cell. We then explored the functional modifications induced in the cell, focusing our attention on the pathophysiological mechanisms that can be involved in the proliferation and death cell processes. Comparative electrophysiological profiling, the analysis of intracellular basal calcium concentration, modification of mitochondria functionality as well as proliferative effects induced by E protein expression will be herein presented.

## 2. Results

### 2.1. Electrophysiological Impact of E Protein Transfection in HEK293 Cells

As the first step, we developed a new different expression vector for SARS-CoV-2 E protein. We used the original Wuhan variant registered in 2020 to obtain a single polycistronic transcript containing a chimeric product with E protein, P2A sequence and eGFP (Figure 1A,B). P2A sequence has the intrinsic ability to cleave itself, separating from E protein and from the eGFP reporter gene [30]. This construct allowed us to get a realistic E protein version, considering its low molecular weight and that little variations in its structure can affect its functionality.

At first, we decided to generate a transfected clone on the HEK293 cell line, a useful model to understand the electrophysiological impact of E protein expression. Transfection with E@ and Mock@ pcDNA3.1 vectors was performed with Lipofectamine^TM^ under the positive selection of the G418 antibiotic (0.5 mg/mL) in the culture medium (see Section 4). Successful stable transfection was obtained for E@ and Mock@ pcDNA3.1 vectors, establishing HEK293@Env and HEK293@Mock cell lines, respectively. The yield of transfection was near to 100%, as demonstrated by fluorescence microscopy analysis (Figure 1C). In agreement with fluorescence analysis, PCR performed on genomic DNA confirmed the presence of the E gene.

Since E protein is reported to act as a viroporin, we verified the functional expression of E protein in HEK293@Env and in HEK293@Mock cells by using the patch-clamp technique (see Section 4). The electrophysiological comparation between HEK293@Mock versus HEK293@Env was recorded using whole-cell dialyzed cell configuration. A linear voltage gradient protocol (ramp) in voltage-clamp mode was applied from −140 mV to +140 mV (V holding −40 mV, Figure 2A) under physiological conditions. An increase in currents along a wide range of potential was observed with a reversion potential estimated at about −59 mV (Figure 2B). The Goldman–Hodgkin–Katz equation applied in our ionic condition displayed a 10 times greater potassium ions selectivity than sodium. The electric capacitances in the two cell lines did not show any significant difference (Figure 2C). The currents promoted by Envelope expression displayed biophysical properties similar to endogenous voltage-dependent potassium current (Kv) reported in HEK293 [31]. These data, however, do not exclude the presence of a very small viroporin aspecific cationic current as reported by other groups [20,21].

### 2.2. Electrophysiological Impact of E Protein Transfection in Human U251 Glioblastoma Cells

Using the same vectors, we established U251@Env and U251@Mock. The yield of these transfections, estimated via fluorescence microscopy, was about 70% (Figure 3A). We then performed the same electrophysiological comparation between U251@Mock and U251@Env in whole-cell perforated configuration to maintain transduction signal mechanisms. As it is possible to see in Figure 3B,C, an increase in outward current was observed at membrane potentials greater than values near the equilibrium potential of potassium ion (E_K_ estimated near to −90 mV by using Nernst equation). The outward current further increased about +50 mV with a characteristic noise after positive potential, previously associated with the big conductance calcium-activated potassium current (BKCa) [29]. This electrophysiological profile resembles the effects derived from an increase in intracellular calcium concentration as observed in glioblastoma cells after calcium mobilizer application, such as histamine [32]. According to this interpretation, we verified the alteration of basal intracellular calcium levels in U251@Mock and U251@Env by using a FURA-2 calcium measurement assay (see Section 4). Interestingly, we observed a slight, but not significant (*p* = 0.15), increase in intracellular basal [Ca^2+^] in U251@Env (Figure 3D), suggesting that the electrophysiological alteration was associated with an increase in calcium-activated potassium currents (IKCa and BKCa) [28,32].

### 2.3. E Protein Expression Induces Mitochondria Depolarization and Promotes U251 Cell Growth

The observation that E protein strongly altered endogenous currents at the cytoplasmic membrane prompted us to investigate the electrophysiological mitochondria properties (ΔΨ_m_) in HEK293- and U251-transfected cells, based on which E protein is mainly expressed in the intracellular membranes. To this purpose, we compared the mitochondrial membrane potential (ΔΨ_m_) by using the nernstian dye TMRM [33] in eGFP+ both HEK293- and U251-transfected cell lines with both vectors (Mock and Env). Interestingly, an inner mitochondrial depolarization was observed in HEK293 and in U251 (Figure 4). Specifically, the ΔΨ_m_ decreased by 21% in U251 and to a similar extent in HEK293.

Since mitochondrial potentials are associated with cell survival, we sought to verify the impact of E protein expression on cell proliferation. As shown in Figure 5, the U251@Env (red) during the first two days displays an increased growth rate respect to U251@Mock until the fourth day, when both cell lines reach full confluence (plateau).

### 2.4. Transcriptional Effects of E Protein Expression in U251 Glioblastoma Cells

The proliferative action of E protein expression was further investigated by comparing the study of the functional expression of genes involved in cell signaling. As shown in Table 1 and Figure 6, we individuated 20 genes that were differently expressed between the two cell lines. Interestingly, among them, we observed a modulation of some genes related to cell proliferation, such as the down-regulation of the pro-apoptotic gene BCL2-associated X (BAX) and an up-regulation of tumor necrosis factor (TNF), previously demonstrated to promote U251 cell proliferation [34]. Further, we observed an up-regulation of WNT1 e WNT6 genes, which are known to inhibit apoptosis. All this gene regulation supports a pro-survival effect of E protein according to the increase in growth rate. Finally, we reported a significant down-regulation of interferon gamma (IFN-γ) and an up-regulation of C-C motif chemokine ligand 5 (CCL5), both genes involved in immunological responses.

A function protein association network analysis (by using the online resource Search Tool for the Retrieval of Interacting Genes—STRING) was then performed. This analysis showed a good interaction between altered genes. Line thickness indicates the strength of data support. The minimum required confidence score was set with a medium confidence of 0.400. The Markov Cluster Algorithm (MCL) was applied [35]. Through this analysis, the 20 dysregulated genes were grouped in four clusters, as shown in Figure 7.

Specifically, we found four clusters: cluster 1 (in red) includes 7 genes that can be related to cytokine activity and, generally, to an inflammatory process. In cluster 2 (in yellow), the 7 genes are strongly correlated in the WNT pathway, which controls differentiation, development and apoptosis. In green (cluster 3), it is possible to see a correlation among the genes that are involved in fatty acid metabolism. Then, in blue, cluster 4 includes 2 genes that are reported to be involved in the apoptosis process.

## 3. Discussion

In this work, we successfully established E protein-expressing U251 cell line (U251@Env) as well as only eGFP-expressing U251 cell line (U251@Mock). E protein expression in U251@Env was confirmed by fluorescence microscopy analysis and biochemically by PCR.

E protein was able to promote mitochondria depolarization in both cellular models. E protein is known to be mainly localized in the intracellular compartment and the alterations of membranes permeability can affect mitochondria functionality. Accordingly, through intracellular localization, such as the inner mitochondrial membrane, we observed a ΔΨ_m_ depolarization [36] in both cell lines. This effect could explain the alteration in cell growth and gene expression observed after E protein expression. Cell proliferation is a possible consequence of the activation of pro-survival pathways. Among the genes that changed as a result of E protein expression, pro-apoptotic genes *BAX* (BCL2 associated X) and *BBC3* (BCL2 binding component 3) were strongly down-regulated (−2.16 and −2.39, respectively). A significant modification was also observed in the expression of the anti-apoptotic *WNT1* and *WNT6* genes, which were both up-regulated (4.90 and 8.47, respectively). It is, therefore, reasonable to suppose that E protein has an anti-apoptotic role, which is pivotal to keep the host cell alive. Apparently, the anti-apoptotic effect induced by E protein is in contrast to ΔΨ_m_ modification, but in the cancer, its relationship with apoptosis is complex, as reported through the Warburg effect [37].

Besides the genes involved in apoptosis control, we also identified other interesting signals that were affected by E protein expression. For instance, CCL5 (C-C motif chemokine ligand 5), a proinflammatory chemokine recruiting leukocytes to the site of inflammation, was up-regulated in U251@Env cells compared to Mock, in accordance with the up-regulation detected in patients with severe COVID-19 [38], suggesting the role of E protein in the cytokine storm. In contrast, E protein seemed to significantly down-regulate IFN-γ, an antiviral chemotactic cytokine, and the transcription factor of other interferons (IRF1), enforcing the idea that E protein tends to create a more protective environment for viral replication. Erythropoietin (EPO) is the main signal driving erythropoiesis and often has been proposed as a possible adjuvant in therapies against COVID-19; among the genes screened, *EPO* showed down-regulation in agreement with the literature [39], possibly being a contributing cause to hypoxia in severe COVID-19. E protein also down-regulated genes of important cell enzymes, such as matrix metalloproteinase 7 (*MMP7*), O-fucosylpeptide 3-beta-N-acetylglucosaminyltransferase (*LFNG*), glutathione-disulfide reductase (*GSR*) and carnitine palmitoyltransferase 2 (*CPT2*). These genes have never been related to SARS-CoV-2 before as far as we know. More interestingly, E protein strongly up-regulated TNF production, one of the most prominent cytokines predicting COVID-19 severity and death [40]. Recently, an over-expression of TNF has been associated with taste dysfunction and, in particular, to taste distortion [41]; E protein could, therefore, be the main cause of COVID-19 ageusia.

A Gene Ontology (GO) analysis was carried out in order to identify biological processes or molecular functions impacted by E protein expression in U251 cells. Thus, we performed a gene enrichment analysis for the gene expression changes induced by E protein, shown in Table 1. Among the 10 reported categories obtained performing a screening with the PANTHER pathway data set, we identified four GO categories that were significantly enriched among these genes (FDR *p* < 0.05) (Table 2). These categories include WNT (*WNT1*, *WNT3A*, *WNT6*, *AXIN2*, *TNF2*, *MMP7*), Alzheimer disease presenilin (*WNT1*, *WNT3A*, *WNT6*, *MMP7*), cadherin (*WNT1*, *WNT3A*, *WNT6*) and Notch (*LFNG*, *HES1*) signaling pathways.

Intracellular calcium studies by FURA-2 highlighted, in the U251 line, a trend of increased intracellular calcium in the U251@Env that could explain the increase in IKCa- and BKCa-activated calcium currents. The increase in intracellular calcium could be due to the following: (i) calcium conduction through viroporin-formed by protein E expression mainly in intracellular membranes (see [21] but also [20]); (ii) the activation of endogenous calcium permeating cation channels such as those of the TRP family, which have been seen to be activated by the WNT pathway [42]; and (iii) disruption in intracellular calcium clearance due to the depolarization of the inner mitochondrial potential [43]. Further studies are needed to understand what signals are involved in the disruption of calcium homeostasis and which calcium potassium currents are promoted upon by protein E.

In conclusion, an evident role of E protein expression can be highlighted in promoting proliferation and inflammation in glioblastoma cell model. Since the U251 glioblastoma cell line, as it was demonstrated, has a good tropism for SARS-CoV-2 (but not for SARS-CoV) [11], our results could explain the neurological disorder given by COVID-19, such as anosmia and ageusia, headache, encephalopathy, demyelinating disorders, stroke and seizures, including Guillain–Barré syndrome [44]. Based on the importance of the ion channel in glioblastoma pathology and therapy [45,46], further experiments are needed to verify the impact of the E protein of SARS-CoV-2 on electrophysiological change with respect to long COVID syndrome and gliomagenesis.

## 4. Materials and Methods

### 4.1. Mock_pcDNA3.1(+)-P2A-eGFP Plasmid Construction

Starting from EnvelopeP_pcDNA3.1(+)-P2A-eGFP (Figure 1A,B), obtained in service (Genscript Biotech Corp, Oxford, UK), the E gene was excised by enzymatic digestion. Briefly, 2 µg of plasmid were digested in a final volume of 50 µL with 1 µL each of NheI and XbaI restriction enzymes (Thermo Fisher Scientific, Rodano, Milan, Italy) and relative buffer (5 µL). Digestion was left at 37 °C for 1.5 h. Proof of successfully linearized Mock_pcDNA3.1(+)-P2A-eGFP production was obtained by agarose gel electrophoresis. Ligation was hence conducted with T4 ligase enzyme (Thermo Fisher Scientific, Rodano, Milan, Italy) to close the plasmid again. The protocol used for ligation was suggested by T4 kit and it was not modified. Proof of successful ligation was given by agarose gel electrophoresis. Two different types of cells were used in this study, namely HEK293, a known and robust model for electrophysiological studies, and U251 in order to get information regarding neurological effects of COVID-19 infection in a glioblastoma model. The E protein gene sequence used was the original Wuhan variant registered in 2020 [UCSC Genome Browser on SARS-CoV-2 Jan. 2020 (NC_045512.2) (wuhCor1)] [47]. We selected this E protein variant since we were interested in exploring long COVID syndrome and, in particular, the neurological effects that were more evident in the pre-Omicron variants. The E protein gene was cloned in service (Genscript Biotech Corp, Oxford, UK) inside a pcDNA3.1(+)-P2A-eGFP vector (E@pcDNA3.1); the full map is reported in Figure 1A,B. In this vector, a single polycistronic transcript from the Cytomegalovirus (CMV) promoter gives a chimeric translation product of E protein, P2A sequence and enhanced green fluorescent protein (eGFP). P2A sequence is a viral protein sequence that has the intrinsic ability to cleave itself; in this way, the final result is the separation of the E protein and eGFP, the report gene. Also, the vector contains the neomycin-resistance (NeoR) gene, which confers resistance to the G418 (also known as geneticin) antibiotic. This vector was particularly suitable for this work for three main reasons. Firstly, the enhancer and promoter of CMV guarantee high protein expression. Secondly, the production of a polyprotein containing both the insert and eGFP guarantees an expression check but, in the meantime, the self-cleaving P2A sequence that separates the two protein sequences allows for protein independence. Thirdly, the expression of the NeoR gene guarantees resistance against the G418 antibiotic, thus allowing for the positive selection of transfected clones. At last, an empty pcDNA3.1(+)-P2A-eGFP vector (Mock@pcDNA3.1) was obtained by excising E sequence with NheI and XbaI restriction enzymes from EnvelopeP_pcDNA3.1(+)-P2A-eGFP.

### 4.2. Cell Culture

All the cells used for this work (U251 and HEK293) were grown in DMEM with high glucose (EuroClone S.p.A., Pero, Milan, Italy) supplemented with 10% FBS (EuroClone S.p.A., Pero, Milan, Italy), 100 IU/mL of penicillin G (EuroClone S.p.A., Pero, Milan, Italy), and 100 µg/mL of streptomycin (EuroClone S.p.A., Pero, Milan, Italy) in an H_2_O-saturated 5% CO_2_ atmosphere at 37 °C and in sterile flasks (Falcon, Corning, Glendale, AZ, USA). Cell confluence was held strictly below 70%, if not differently specified.

### 4.3. Stable Transfections

U251 or HEK293 cells were seeded in a 6-well multi-well at a density of 100,000 cells for each well. The day after, transfection was performed with the Lipofectamine 3000 (Thermo Fisher Scientific, Rodano, Milan, Italy) protocol. After 48 h, the transfection antibiotic Geneticin 418 (Sigma-Aldrich, St. Louis, MO, USA) was applied to the culture at the concentration of 0.5 mg/mL. The antibiotic was removed from the cell culture after complete recovery. The yield of transfection was estimated by fluorescent microscopy (Axiozoom V16, Zeiss, Jena, Germany, equipped with Axio-105 color camera) using a GFP fluorescence filter and bright field to quantify the percentage of eGFP-positive cells.

### 4.4. PCR and rtPCR

Both RNA and DNA from transfected U251 or HEK293 cells were extracted using the Trizol (Invitrogen, Waltham, MA, USA) method according to the manufacturer’s instructions. PCR on genomic DNA was led with the Real-Time RotorGene 100 and a Taq Polimerase kit by Thermo Fisher Scientific, following the manufacturer’s instructions. For the detection of E protein gene, the following primers were used:

Forward: ATGTACTCATTCGTTTCGGAAGAG

Reverse: GACCAGAAGATCAGGAACTCTAG.

For gene expression, RNA was reverse-transcribed into first-strand cDNA using the Kit RT2 First Strand and analyzed with the PAHS-014Z RT2 Profiler PCR Array (Qiagen, Hilden, Germany) using RT2 SYBR Green ROX FAST Mastermix as a reagent (Qiagen, Hilden, Germany) and Real-Time RotorGene 100 (Qiagen, Hilden, Germany) following the manufacturer’s instructions. Results were expressed as fold change comparing Mock- and E protein-expressing cells. The cutoff was fixed at the 35th cycle.

### 4.5. Patch Clamp Recordings

Whole-cell perforated and dialyzed configurations were used for electrophysiological recordings from HEK293 and U251 cells. Currents were amplified with a HEKA EPC-10 amplifier and analyzed with the PatchMaster and Origin 4.1 software. For online data collection, currents were filtered at 3 kHz and sampled at 40 μs/point. Membrane capacitance measurements were made by using the transient compensation protocol of PatchMaster. In whole-cell recordings under physiological conditions, we used a modified Ringer external solution containing (in mM) 140 NaCl, 5 KCl, 2 CaCl_2_, 2 MgCl_2_, 5 MOPS, 10 glucose (pH 7.4) and the pipette solution containing (in mM) 155 KCl, 1 MgCl_2_, 5 MOPS, and 1 EGTA-K (pH 7.20). Electrical access ranging between 10 and 20 MΩ, after cell membrane breaking and the desired free Ca^2+^ concentration, [Ca^2+^]_i_, was obtained by adding varying amount of CaCl_2_ (calculated with the webmax software Webmaxc Standard available online: https://somapp.ucdmc.ucdavis.edu/pharmacology/bers/maxchelator/webmaxc/webmaxcS.htm accessed on 10 January 2024) to the bathing solution. All the chemicals used were of analytical grade. In the perforated-patch configuration, we used an external solution containing (in mM) 106.5 NaCl, 5 KCl, 2 CaCl_2_, 2 MgCl_2_, 5 MOPS, 20 glucose, and 30 Na-gluconate (pH 7.25) and a pipette solution containing (in mM) 57.5 K_2_SO_4_, 55 KCl, 5 MgCl_2_, 10 MOPS and 20 glucose (pH 7.20). Electrical access to the cytoplasm was achieved by adding amphotericin B (200 μM) to the pipette solution. Access resistances ranging between 15 and 25 MΩ were achieved within 10 min following seal formation and were actively compensated to ca. 50%. All experiments were carried out at room temperature (18–22 °C). Data are presented as means ± SE. Statistical differences between experimental groups were verified by using the *t*-test and considering the level of significance (*p*) of 0.05.

In order to avoid gap–junction interference, only not-coupled cells were considered. To distinguish the eGFP+ cells for patch clamp measurements, plates were mounted on an inverted fluorescent microscope (Axiovert 200, Zeiss, Jena, Germany) with GFP and bright-field channels and a color camera (True Chrome II s, TiEsseLab, Milan, Italy).

### 4.6. Calcium Measurement Assay

U251 cells expressing E protein or Mock were seeded in 35 mm Petri dish plates (Thermo Fisher Scientific) at a density of 50,000 cells for plate. After two days, cells were mounted under an immersion fluoresce microscope (AxioExaminer, Jena, Germany) and under a gravity-drive perfusion system, with tubing connected to a final tip of 100- to 200-µm diameter focally oriented onto the field of interest. As eGFP interferes with ratiometric fluorescent dye FURA-2 AM, we created a modified version of the standard protocol for calcium measurement as follows. After the first frame was taken (T = 0, used as baseline), 3 µM of FURA-2 AM (Sigma-Aldrich) were added to the plate and incubated for 40 min (an image was snapped every 5 min), then it was suddenly removed with a perfused Ringer solution and frames were taken every minute. Cells were monitored for over 10 min. The estimation of intracellular free Ca^2+^ concentration for eGFP+ cells was reported as change of the ratio between fluorescence emission at 510 nm, obtained with 340 and 380 nm excitation wavelengths almost simultaneously emitted (optical filters and dichroic beam splitter were from Lambda DG4, Sutter Instruments, Novato, CA, USA), after dye removal. The absolute values of 340 and 380 channels were subtracted to T = 0 values.

### 4.7. ΔΨ_m_ Estimation

For the evaluation of membrane potential ΔΨ_m_, the Tetra Methyl Rhodamine Methyl ester (TMRM) nernstian fluorescent stain (Sigma-Aldrich) was used. The dye was used according to the manufacturer’s instructions. All experiments were carried out contemporaneously on U251@Env and U251@Mock cells and on HEK293@Env and HEK293@Mock cells. Cells were seeded the day before the experiment in 35 mm Petri dish plates (100,000 cells for plate); the day after, cells were incubated in DMEM with 30 nM of TMRM for 20 min. At the end of the incubation time, plates were washed with phosphate-buffered saline and analyzed by fluorescent microscopy AxioExaminer (Zeiss) using the Rhodamine filter. Results are expressed as mean intensity normalized (±SE) for Mock cells.

### 4.8. Growth Curve Assessment

U251 cells, both E protein-expressing and Mock, were seeded in 6-well plates at a density of 30,000 cells per well. Every day for four days, cells underwent trypsinization and were counted with the Bürker chamber after being diluted 1:1 with the vital dye Tripan Blue (Sigma-Aldrich) in order to discard dead cells from the count. The curves were then elaborated with the software Origin 6.1 (OriginLab Corporation, Northampton, MA, USA); each point of the curve is the result of three independent counts (±SE).

### 4.9. Gene Ontology and Protein Interaction Network Analysis

To identify which biological process was more closely associated with the gene expression modified by E protein in U251 cells, we used the Gene Ontology Resource available online by the GO Consortium https://geneontology.org/ accessed on 14 May 2024. The analysis performed is the PANTHER Overrepresentation Test (Released 20240226), using Homo Sapiens (all genes in the database) as the reference list and the list of genes modified by E protein expression as the analyzed list. The process initially involves uploading the list of genes to be analyzed with common gene identifiers. Then, the list undergoes analysis using, as the annotated data set, the PANTHER pathway, which consists of over 177, primarily signaling, pathways, each with PANTHER subfamilies and protein sequences mapped to individual pathway components. The raw *p*-value was determined by Fisher’s exact test, and FDR (False Discovery Rate) was used as a correction to define the probability that the number of the observed genes in this category occurred by chance (randomly). Subsequently, we obtained a table with the classified genes and their fold enrichment values.

To build the protein interaction network, we used the online database Search Tool for the Retrieval of Interacting Genes (STRING) v. 9.1 (http://string-db.org accessed on 14 May 2024) [48]. The interactions include direct (physical) and indirect (functional) associations derived from four sources: genomic context, high-throughput, co-expression and prior knowledge. Line thickness indicates the strength of data support. The minimum required confidence score was set with a medium confidence of 0.400. The Markov Cluster Algorithm (MCL) was applied [35].

## Figures and Tables

**Figure 1 ijms-25-06669-f001:**
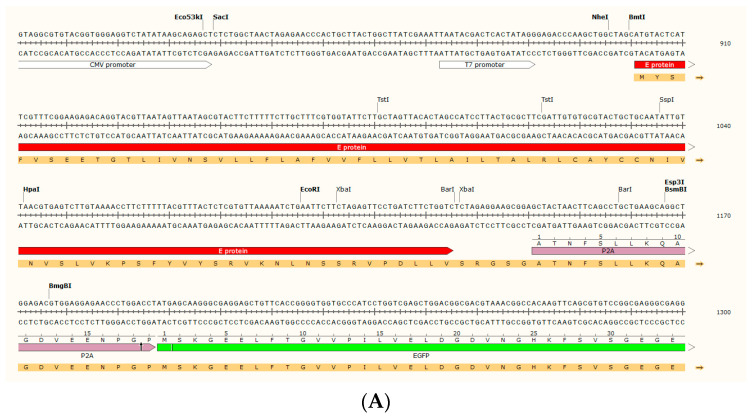
Transfection of HEK293 cells with E protein. (**A**) Plasmid sequence detail, enlarged on the zone of E sequence (in red), P2A (in purple) and eGFP (in green). The black arrow inside the P2A sequence indicates the point of the self-cleavage of the P2A protein. The amino acid sequence of the transcript is shown in orange. (**B**) EnvelopeP_pcDNA3.1(+)-P2A-eGFP full map. E sequence is indicated in red, eGFP sequence is indicated in green. * means that the restriction enzyme cleavage is blocked by Dcm methylation. For additional details see www.genscript.com. (**C**) Example fields (in the bright field, GFP channel and merge) of HEK293@Env established cell line. Cells containing the plasmid express eGFP (transfection yield 100%). Scale bar: 100 µm.

**Figure 2 ijms-25-06669-f002:**
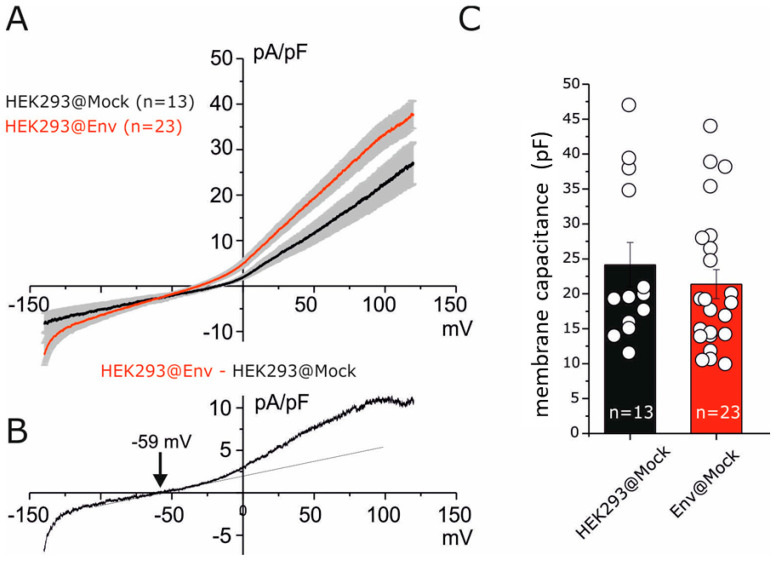
Electrophysiological effects of E protein expression in HEK293 cells. (**A**) Mean I–V relationship of HEK293@Mock (black, n = 13) and HEK293@Env (red, n = 23). The currents are reported as current density obtained by the ratio of single-cell currents with electric capacitance (pA/pF). (**B**) Current density difference obtained by the numerical subtraction of red and black traces displayed in (**A**). The dash line represents the linear conductance that described ohmic behavior at negative potential until −50 mV. Note the outward deviation up to −50 mV, indicating the voltage dependance similar to endogenous currents and ascribed to the voltage-dependent potassium current. (**C**) Mean capacitances of the cells reported in the recordings of (**A**).

**Figure 3 ijms-25-06669-f003:**
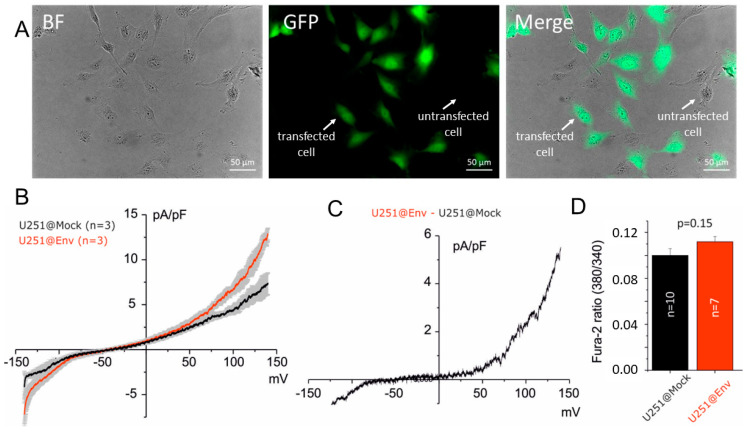
Electrophysiological effects of E protein expression in U251 cells. (**A**) Example fields (in bright-field, GFP channel and merge) of U251@Env established cell line. Cells containing the plasmid express eGFP and therefore are discriminable from untransfected cells with fluorescence microscope and GFP channel (middle), as indicated by the white arrows. Transfection yield: 70%. Scale bar: 50 µm. (**B**) I–V relationship of U251@Mock (black, n = 3) and U251@Env (red, n = 3). The impedance was normalized for single cell capacitance. The current is reported as current density obtained by the ratio of single cell currents with electric capacitance (pA/pF). (**C**) Current density difference obtained by the numerical subtraction of red and black trace displayed in B. Note the increase in the outward noisy current after +50 mV according to BKCa currents. (**D**) Effects of E protein expression on calcium basal levels in U251@Mock (black) and U251@Env (red) recorded after 5 min of Fura-AM incubation (see Section 4 for details). Note the trend of an increase in intracellular calcium levels in U251@Env compared to U251@Mock (*p* = 0.15).

**Figure 4 ijms-25-06669-f004:**
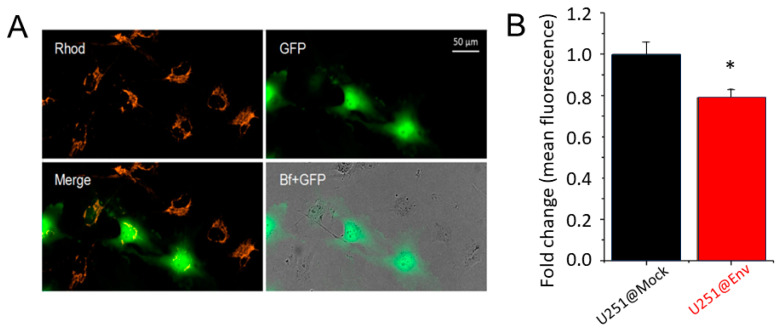
Effects of E protein expression on mitochondria functionality. (**A**) Example fields (in Rhodamine and GFP channels, Merge and Merge of GFP and bright-field channels) of U251@Env after TMRM (observable in Rhodamine channel) incubation. (**B**) Mean ΔΨ_m_ measurement in U251@Mock (black) and U251@Env (red), displayed after normalization. * *p* < 0.05 using unpaired *t*-Test analysis.

**Figure 5 ijms-25-06669-f005:**
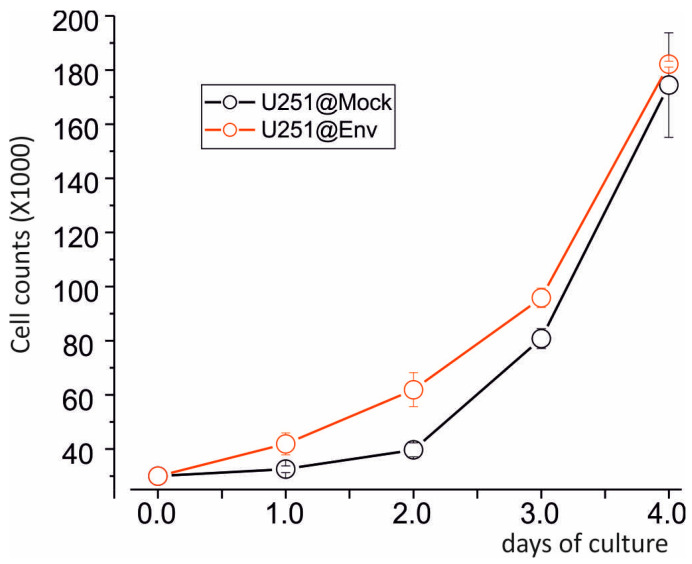
Effects of E protein expression on proliferation and mitochondria functionality. Mean growth curves of U251@Mock (black) and U251@Env (orange).

**Figure 6 ijms-25-06669-f006:**
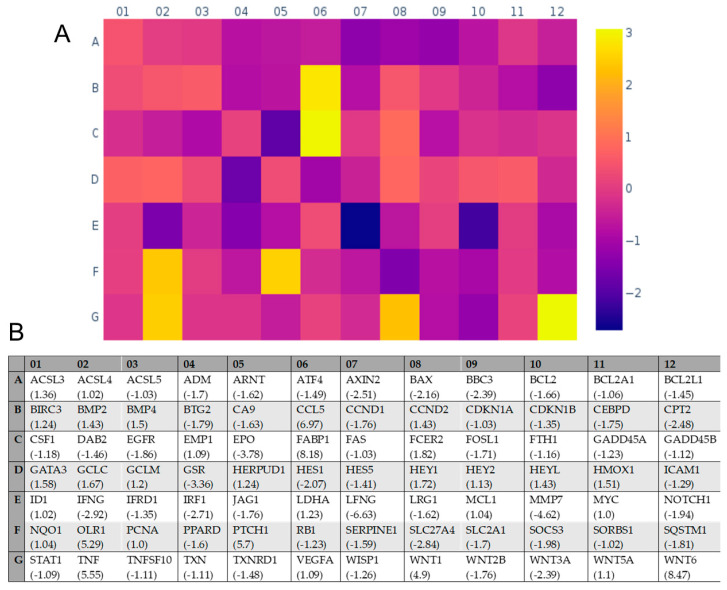
Effect of E protein expression on gene expression. (**A**) Diagram reporting the complete results of the gene array screened by rtPCR. The color legend on the right indicates the fold change. (**B**) Map indicating the genes name of every position and the fold of change (in brackets).

**Figure 7 ijms-25-06669-f007:**
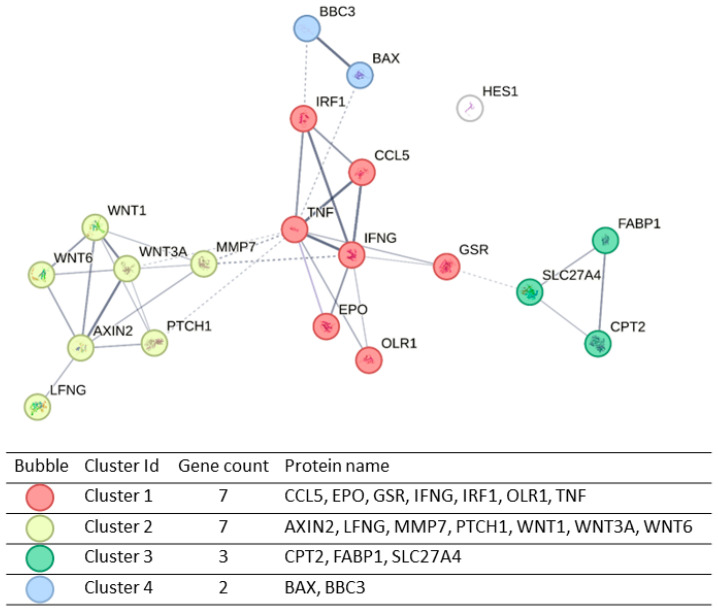
STRING analysis of gene impacted by E protein expression in U251 cells. White bubble indicates a gene that is not included in any cluster.

**Table 1 ijms-25-06669-t001:** Genes modulated by E protein expression.

Gene Name ^1^	Mock Average ΔCT ^2^	U251@Env Average ΔCT ^2^	Fold Change
*LFNG*	8.50	11.23	−6.63
*MMP7*	8.26	10.47	−4.62
*EPO*	11.58	13.50	−3.78
*GSR*	8.74	10.49	−3.36
*IFNG*	10.72	12.27	−2.92
*SLC27A4*	5.73	7.24	−2.84
*IRF1*	5.72	7.16	−2.71
*AXIN2*	10.73	12.06	−2.51
*CPT2*	9.25	10.56	−2.48
*BBC3*	9.23	10.49	−2.39
*WNT3A*	12.78	14.04	−2.39
*BAX*	3.34	4.45	−2.16
*HES1*	3.79	4.84	−2.07
*WNT1*	15.12	18.83	4.90
*OLR1*	15.12	12.72	5.29
*TNF*	15.12	12.65	5.55
*PTCH1*	14.87	12.36	5.70
*CCL5*	15.12	12.32	6.97
*FABP1*	15.12	12.09	8.18
*WNT6*	15.12	12.04	8.47

^1^ According to HUGO Gene Nomenclature Committee. *LFNG*: O-fucosylpeptide 3-beta-N-acetylglucosaminyltransferase; *MMP7*: matrix metallopeptidase 7; *EPO*: erythropoietin; *GSR*: glutathione-disulfide reductase; *IFNG*: interferon gamma; *SLC27A4*: solute carrier family 27 member 4; *IRF1*: interferon regulatory factor 1; *CPT2*: carnitine palmitoyltransferase 2; *BBC3*: BCL2 binding component 3; *BAX*: BCL2 associated X; *HES1*: hes family bHLH transcription factor 1; *OLR1*: oxidized low density lipoprotein receptor 1; *TNF*: tumor necrosis factor; *PTCH1*: patched1; *CCL5*: C-C motif chemokine ligand 5; *FABP1*: fatty acid binding protein 1. ^2^ Each gene expression was evaluated by the comparative cycle threshold (ΔCT) method.

**Table 2 ijms-25-06669-t002:** Gene ontology analysis on genes modulated by E protein expression.

PANTHER Pathways	# ^1^	# ^2^	Expected ^3^	Fold Enrichment ^4^	Raw *p* Value ^5^	FDR ^6^
WNT signaling pathway	309	6	0.32	19.04	4.88 × 10^−7^	7.86 × 10^−5^
Alzheimer disease-presenilin pathway	126	4	0.13	31.13	7.38 × 10^−6^	3.96 × 10^−4^
Cadherin signaling pathway	164	3	0.17	17.94	5.94 × 10^−4^	2.39 × 10^−2^
Notch signaling pathway	44	2	0.04	44.57	9.13 × 10^−4^	2.94 × 10^−2^
Apoptosis signaling pathway	119	2	0.12	16.48	6.47 × 10^−3^	1.74 × 10^−1^
CCKR signaling map	172	2	0.18	11.40	1.31 × 10^−2^	2.64 × 10^−1^
Angiogenesis	171	2	0.17	11.47	1.30 × 10^−2^	2.99 × 10^−1^
Hedgehog signaling pathway	20	1	0.02	49.03	2.02 × 10^−2^	3.62 × 10^−1^
Interferon-gamma signaling pathway	30	1	0.03	32.69	3.02 × 10^−2^	4.42 × 10^−1^
Inflammation mediated by chemokine and cytokine signaling pathway	262	2	0.27	7.49	2.89 × 10^−2^	4.65 × 10^−1^

^1^ Number of genes in the reference list that map to this category. ^2^ Number of genes modified by E protein expression that map to this category. ^3^ Expected number of genes in the analyzed list for this category, based on the reference list. ^4^ Fold enrichment of the genes observed in the analyzed list over the expected. ^5^ Raw *p*-value as determined by Fisher’s exact test. ^6^ False Discovery Rate, that is, the probability that the number of observed genes in this category occurred by chance (randomly).

## Data Availability

The original contributions presented in the study are included in the article, further inquiries can be directed to the corresponding authors.

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
