# Peer review of "Electrophysiological Impact of SARS-CoV-2 Envelope Protein in U251 Human Glioblastoma Cells: Possible Implications in Gliomagenesis?"

_ijms, 2024, doi:10.3390/ijms25126669_

Round 1

Reviewer 1 Report

Comments and Suggestions for Authors

In this study, PI describe Electrophysiological impact of SARS-CoV-2 Envelope protein in both HEK293 and U251 human glioblastoma cells. Data provided based on the importance of ion channel and conclude an evident role of E protein expression can be highlighted in promoting proliferation and inflammation in U251 cell line model. It appears very interesting. However, below have some comments:

1.     The rationale seems not sufficient due to the main target cell of SARS-CoV-2 better on lung cells, such as lung carcinoma epithelial cells, A549 vs Human embryonic kidney 293 cells

2.     U251 and HEK293 are different cell lines, do these two cell lines possess the same receptor for E protein for entry?

3.     Same as above question: please make more clear description on Figure 1 legend of: You seem to mix together with both U251 and HEK293@Env on transfection on this Figure. (Tansfection of U251 cells with E protein. A) Plasmid sequence detail, zoomed on the zone 148 of E sequence (underlined in red), P2A (in purple) and eGFP (in green). The black arrow inside P2A 149 sequence indicates the point of self-cleavage of P2A protein. B) EnvelopeP_pcDNA3.1(+)-P2A-eGFP 150 full map. E sequence is indicated in red, eGFP sequence is indicated in green. C) Example photo (in 151 bright-filed, GFP channel and merge) of HEK293@Env established cell line).

4.     Line 218, EK293 (data not shown), HEK293 misspelling?

5.     Line 254, HEK297, HEK293 misspelling?

Author Response

  1. Summary

Thank you very much for taking the time to review this manuscript. Please find the detailed responses below and the corresponding revisions/corrections highlighted in the re-submitted file.

  1. Point-by-point response to Comments and Suggestions for Authors

Comment 1: The rationale seems not sufficient due to the main target cell of SARS-CoV-2 better on lung cells, such as lung carcinoma epithelial cells, A549 vs Human embryonic kidney 293 cells.

Response 1: Thank you for pointing this out. We clearly agree with your observation regarding the preferential trophism shown by SARS-CoV-2 virus toward lung cells, accountable for the known severe symptomatology associated (i.e. interstitial pneumonia). However, our study focuses on the implication of COVID-19 disease in pathologies of the central nervous system and, in particular, on a possible inference on gliomagenesis. For this reason, we selected as model the cellular line U251, that are obtained from a human glioblastoma astrocytoma and that represent a commonly used experimental model of glioblastoma. In particular, among the different types of glioblastoma models, these cells are very suitable for proliferation studies, that represent one of the major aims of the research presented in this manuscript.

To underline this, in the manuscript we added a short motivation explaining the main characteristics and the reason for which we decided to use this cell line.

“U251 cells were selected as cell line on which to perform the studies because they represent a robust and established model of human glioblastoma. Moreover, literature data suggest that SARS-CoV-2 is able to replicate in this cell line.”

Comment 2: U251 and HEK293 are different cell lines, do these two cell lines possess the same receptor for E protein for entry?

Response 2: The technique we perform to induce E protein expression in the different cell lines is a transfection. In this method there is not a particular receptor to allow the plasmide vector to enter the cell, but in the procedure lipofectamine is used to ensure the access of the E protein coding DNA in the cell. In detail, lipofectamine is a cationic lipid capable of binding exogenous DNA, encapsulating it within small artificial vesicles, which, once in contact with the plasma membrane of the cells to be transfected, fuse with it and carry it into the cytoplasm.

Starting from these considerations, we can suppose that there are not mechanistic differences among the two cell lines to be transfected. However, it is worthy to note that, as reported in the manuscript, HEK293 are transfected almost in 100% yield, while U251 are transfected in 70% yield. 

Comment 3: Same as above question: please make more clear description on Figure 1 legend of: You seem to mix together with both U251 and HEK293@Env on transfection on this Figure. (Transfection of U251 cells with E protein. A) Plasmid sequence detail, zoomed on the zone of E sequence (underlined in red), P2A (in purple) and eGFP (in green). The black arrow inside P2A sequence indicates the point of self-cleavage of P2A protein. B) EnvelopeP_pcDNA3.1(+)-P2A-eGFP full map. E sequence is indicated in red, eGFP sequence is indicated in green. C) Example photo (in bright-filed, GFP channel and merge) of HEK293@Env established cell line).

Response 3: Thank you for notice this, there was a mistake in the legend. The current legend is:

“Transfection of HEK293 cells with E protein. A) Plasmid sequence detail, enlarged on the zone of E sequence (in red), P2A (in purple) and eGFP (in green). The black arrow inside P2A sequence indicates the point of self-cleavage of P2A protein. B) EnvelopeP_pcDNA3.1(+)-P2A-eGFP full map. E sequence is indicated in red, eGFP sequence is indicated in green. C) Example fields (in bright-field, GFP channel and merge) of HEK293@Env established cell line. Cells containing the plasmid express eGFP (transfection yield 100%). Scale bar: 100 µm.”

Comment 4: Line 218, EK293 (data not shown), HEK293 misspelling?

Response 4: Yes, there was a misspelling, we corrected EK293 with HEK293.

Comment 5: Line 254, HEK297, HEK293 misspelling?

Response 5: Yes, there was a misspelling, we corrected HEK297 with HEK293.

Reviewer 2 Report

Comments and Suggestions for Authors

In the manuscript titled "Electrophysiological impact of SARS-CoV-2 Envelope protein in U251 human glioblastoma cells: possible implications in gliomagenesis?" by a team of investigators led by Monarca and Fioretti, the transcriptional effect of E protein on the signaling pathways is described, and the impact of E protein expression in gliomagenesis is explored through electrophysiological means.

Although there were several interesting points made here, this manuscript suffers from several weaknesses that may preclude acceptance for publication. 

1. The author need to describe the relationship between E protein and SARS-CoV-2 in a few sentences in the abstract. 

2. Figure 1A and B, Figure 6 have a low resolution, and thus, need improvement.

3. Control needs to be shown for the untransfected group in Figure 1C and Figure 3A.

4. At least two glioblastoma cell lines need to be included to study the function of the E protein in section 2.3. Dose the E protein affect the apoptosis?

5. The author needs to perform pathway analysis or gene enrichment analysis for the gene expression changes induced by E protein in Figure 6 and Table 1. The discussion section should describe the analysis performed. GSE214150 and GSE210497 are likely tools used for analyzing the GO analysis and gene enrichment analysis for the E protein.

Author Response

  1. Summary

Thank you very much for taking the time to review this manuscript. Please find the detailed responses below and the corresponding revisions/corrections highlighted in the re-submitted file.

  1. Point-by-point response to Comments and Suggestions for Authors

Comment 1: The authors need to describe the relationship between E protein and SARS-CoV-2 in a few sentences in the abstract. 

Response 1: Thank you for suggestion. We added a short explanation of this relationship in the abstract as suggested.

Comment 2: Figure 1A and B, Figure 6 have a low resolution, and thus, need improvement.

Response 2: We agree with you. To overcome this problem, we changed Figure 1 in order to make possible to catch all the desired information. We also improved Figure 6, in particular we edited figure 6B, with the map of the genes and the fold of change, now clearly readable.

Comment 3: Control needs to be shown for the untransfected group in Figure 1C and Figure 3A.

Response 3: Thank you for suggestion. In Figure 3A (U251 cells) the transfection yield is 70%, therefore in the same field we can see also untransfected cells (that are present in the BF and in the merge, but not are not GFP sensitive). Therefore, we edited Figure 3A, adding arrows indicating transfected and untransfected cells in the different fields. In Figure 1C (HEK293) the transfection yield is near 100%, therefore it is not possible to evidence the untransfected cells. We modify the caption of the Figure 1C to explain this.  

Comment 4: At least two glioblastoma cell lines need to be included to study the function of the E protein in section 2.3. Dose the E protein affect the apoptosis?

Response 4: Thanks for pointing that out. Unfortunately, we do not have data on another glioblastoma cell line with similar tropism for Sars-Cov-2 to generalize the function of the E protein in apoptosis. However, in order to increase information on the effects of E protein expression on U251 apoptosis we performed a STRING analysis on the transcriptomic data.  We specifically ran a protein association network analysis function (STRING, from the online resource https://string-db.org/) and have included a new table and panel in Figure 7, highlighting the pathways which are involved in the apoptosis process. Moreover, the Discussion section was edited in order to make clearer the effects that we observed after E protein expression.

A function protein association network analysis (by using the online resource Search Tool for the Retrieval of Interacting Genes - STRING) was then performed. This analysis showed a good interaction between these GO‐categories, indicating a regulation network between the impacted genes. Line thickness indicates the strength of data support. The minimum required confidence score was set with a medium confidence of 0.400. Markov Cluster Algorithm (MCL) was applied [40]. Through this analysis, the 20 dysregulated genes were grouped in 4 clusters, as shown in Figure 7.

Figure 7. STRING analysis of gene impacted by E protein expression in U251 cells.

Cluster 1 (in red) includes 7 genes that can be related to cytokine activity and, generally, to inflammatory process. In cluster 2 (in yellow) the 7 genes that are strongly correlated in the WNT pathway, which controls differentiation, development and apoptosis. In green (cluster 3) it is possible to see a correlation among genes that are involved in fatty acid metabolism. Then, in blue, cluster 4 includes 2 genes that are reported to be involved in apoptosis process.

Comment 5: The author needs to perform pathway analysis or gene enrichment analysis for the gene expression changes induced by E protein in Figure 6 and Table 1. The discussion section should describe the analysis performed. GSE214150 and GSE210497 are likely tools used for analyzing the GO analysis and gene enrichment analysis for the E protein.

Response 5: Thank you for suggestion. We performed a Gene Ontology analysis (by the Gene Ontology Resource available online by the GO Consortium https://geneontology.org/) and added a new Table in the discussion section, along with a new paragraph in the Method section to describe the analysis performed. GSE214150 and GSE210497 are nice tools, but we decided not to use them, since our experimental conditions are significantly different from that used to define these data sets.

Reviewer 3 Report

Comments and Suggestions for Authors

1. Summary

In this manuscript, authors addressed the Envelope protein of SARS-CoV-2 virus show eletrophysiological impact to enhancing proliferation and inflammation in gliblastoma cell model by investigating it in U251 and HEK297 models.The total logic and experiment design are relatively good. The description of the object, process with relative explain, detailed information,  results with interpretation are clear and well organized. The methods used for both experiments and statistical analysis are well listed. My comments is to accept this manuscript after minor revision. The detailed comments are as follows: 

2. Detailed comments

1. Please double check the typos in the manuscript. The examples can be:

1. Page 1, Line 19: Do you mean "There are xxxx" in "The are several symptoms xxxxx"? 

2. Page 1, Line21: It should be "pre-Omicron".

3. Page 7, Line 178: "2.2 Electrophysiological xxxxx cellss" to "2.2 Electrophysiological xxxxx cells"

2. Please improve the figures and their quality: 

1. Please use the same number mark, like all A, B in Fig 1-4, or a), b) in Fig 6. And check on the format of the letters like size.

2. Figure1.B: The letters are not readable in the manuscript.

3. Figure 6b: the content in b) is not readable 

Comments on the Quality of English Language

The general descriptions are in good condition. There are some minor typos and grammar errors needs to be corrected.

Author Response

  1. Summary

Thank you very much for taking the time to review this manuscript. Please find the detailed responses below and the corresponding revisions/corrections highlighted in the re-submitted file.

  1. Point-by-point response to Comments and Suggestions for Authors

Comment 1: Please double check the typos in the manuscript. The examples can be:

  1. Page 1, Line 19: Do you mean "There are xxxx" in "The are several symptoms xxxxx"? 
  2. Page 1, Line21: It should be "pre-Omicron".
  3. Page 7, Line 178: "2.2 Electrophysiological xxxxx cellss" to "2.2 Electrophysiological xxxxx cells"

Response 1: Thank you very much for notice this. We have corrected the errors you pointed out to us, and we have also carefully checked the rest of the text to find and correct other typos.

Comment 2: Please improve the figures and their quality: 

  1. Please use the same number mark, like all A, B in Fig 1-4, or a), b) in Fig 6. And check on the format of the letters like size.
  2. Figure1.B: The letters are not readable in the manuscript.
  3. Figure 6b: the content in b) is not readable

Response 2: Your comments are correct. We have uniformed the number mark and the letter size for all the figures. Figure 1B was enlarged to be better understood. We also edited figure 6B, with the map of the genes and the fold of change, now clearly readable.

Round 2

Reviewer 2 Report

Comments and Suggestions for Authors

The authors' response in the revised manuscript, in reaction to the initial critiques, adequately addressed the issues raised. The resulting manuscript has significantly improved and can be accepted in its current form.